# The feeder system of the Toba supervolcano from the slab to the shallow reservoir

Ivan Koulakov[1,2], Ekaterina Kasatkina[1,2], Nikolai M. Shapiro[3,4], Claude Jaupart[3], Alexander Vasilevsky[1,2], Sami El Khrepy[5,6], Nassir Al-Arifi[5] & Sergey Smirnov[7,8]

The Toba Caldera has been the site of several large explosive eruptions in the recent geological past, including the world's largest Pleistocene eruption 74,000 years ago. The major cause of this particular behaviour may be the subduction of the fluid-rich Investigator Fracture Zone directly beneath the continental crust of Sumatra and possible tear of the slab. Here we show a new seismic tomography model, which clearly reveals a complex multilevel plumbing system beneath Toba. Large amounts of volatiles originate in the subducting slab at a depth of $\sim 150\,km$, migrate upward and cause active melting in the mantle wedge. The volatile-rich basic magmas accumulate at the base of the crust in a $\sim 50,000\,km^3$ reservoir. The overheated volatiles continue ascending through the crust and cause melting of the upper crust rocks. This leads to the formation of a shallow crustal reservoir that is directly responsible for the supereruptions.

[1] Trofimuk Institute of Petroleum Geology and Geophysics, SB RAS, Prospekt Koptyuga, 3, 630090 Novosibirsk, Russia. [2] Novosibirsk State University, Pirogova 2, 630090, Novosibirsk, Russia. [3] Institut de Physique du Globe de Paris, Sorbonne Paris Cité, CNRS (UMR 7154), 1 rue Jussieu, 75238 Paris, Cedex 5, France. [4] Institute of Volcanology and Seismology, FEB RAS, 9 Piip Boulevard, Petropavlovsk-Kamchatsky, 693006 Kamchatsky Region, Russia. [5] King Saud University, PO Box 2455, Riyadh 11451, Saudi Arabia. [6] National Research Institute of Astronomy and Geophysics, NRIAG, 11421, Helwan, Egypt. [7] Sobolev Institute of Geology and Mineralogy SB RAS, Prospekt Koptyuga, 3, 630090 Novosibirsk, Russia. [8] Tomsk State University, 36 Lenin Avenue, 634050 Tomsk, Russia. Correspondence and requests for materials should be addressed to I.K. (email: KoulakovIY@ipgg.sbras.ru).

The Toba Caldera, located in northern Sumatra, has been the site of several large explosive eruptions over the past million years. The most recent Toba supereruption, ~74,000 years ago, is considered to be the largest terrestrial volcanic eruption of the Pleistocene[1,2]. This eruption ejected an enormous volume of material, with the dense rock equivalent estimated between 2,800 and 5,300 km[3] (refs 1,3). The scale of the eruption significantly affected the global biosphere and climate[4,5], although some specialists[6] suggest that commonly accepted estimates of the catastrophic consequences of the Toba supereruptions are overestimated. In any case, if such an eruption occurred in modern times, it would drastically alter human life. Therefore, it is critical to understand the functioning of the magmatic system that periodically produces such unusually voluminous eruptions.

The Toba Caldera is one of the volcanic complexes of the Sunda Arc, where the Indo-Australian Plate subducts obliquely at a rate of 56 mm per year[7]. In different subduction zones, many traces of caldera-forming eruptions have been recorded[8]; however, the intensity and repeatability of the eruptions at Toba make it unique. The topography of the Toba area is characterized by a large uplift with an average elevation of ~1,500 m and lateral dimensions of 220 × 100 km (area indicated by blue dotted line in Fig. 1). This area is transected by the Great Sumatran Fault Zone (GSFZ), which can be clearly identified on the topographic map. Abundant evidence of recent volcanic activity, such as circular caldera structures and cinder cones, is also clearly visible on the topographic map.

In the context of the Toba volcanic activity, several important questions are actively debated: why did several large eruptions occur in approximately the same location in the Toba Caldera area? Why were the supereruptions followed by long periods of quiescence? Is the magma system beneath Toba active and can we expect a new supereruption of Toba in the near future? In this study, we analyse several types of observations, such as topographic/bathymetric maps, geoid transformations, seismicity distributions, and regional and local seismic tomography, and attempt to shed light on some of the above-mentioned questions. Our results show that the anomalous style of magma production beneath the Toba Caldera is primarily caused by perturbations on the slab associated with the subducting fracture zone. In the tomography model, we can identify traces of the magma and volatile pathways in the mantle and two large magma reservoirs on the base of the Moho interface and in the upper crust, which are the major elements responsible for episodic occurrence of supereruptions.

## Results

### Analysis of the relief and gravity data

The present-day activity of the Toba supervolcano is fundamentally determined by its geographic location and geological setting. Figure 2 summarizes the locations of the main volcano-related structures and the Great Sumatran Fault, identified from the topographic map and from literature[1,2]. The locations of the three most recent calderas, formed within the last million years[2], are also shown in Fig. 2.

Consideration of topographic/bathymetric maps may provide some insight into the processes causing the supervolcanism. As shown in Fig. 3a, the Toba Caldera is located above the prolongation of the Investigator Fracture Zone (IFZ), which is a 2,500-km-long transform zone in the Indian Ocean. The IFZ separates the younger northwestern segment of the oceanic plate, which has an age of ~40 Ma, from the southeastern segment, which is ~15 Ma older[9]. On the outer rise of the subducting plate, the IFZ splits into several parallel ridges with elevations up to 1,500 m above the surrounding seafloor. The bathymetric map shows that the morphology of the accretionary complex at the

junction between the IFZ and the Sunda Trench is considerably different compared with the other segments of the trench, which may be explained by stronger shortening of the overriding plate[10] in this area.

Figure 3b presents a map of the longitude-directed derivative of the geoid model EIGEN-6C4 (ref. 11). This transformation is effective for revealing latitude-oriented linear structures related to the IFZ. These features represent deep density variations beneath the fracture zone, some of which are continuous across the trench. The possibility of using gravity data to detect the signatures of crustal features in different subduction zones has recently been investigated[12]. The most prominent continuation of the IFZ to the onshore Sumatran area is observed in the fracture line directed towards Toba. In Fig. 3b, we also plot the slab-related earthquakes identified by local seismic networks. Most of the seismicity defines a linear dipping structure that extends from the lineaments of the IFZ (see also Supplementary Fig. 1). Same earthquakes, but on a larger scale, are presented in Fig. 2, which shows that the narrow seismicity zone associated with the subduction of the IFZ is located directly beneath the southern part of the Toba Caldera.. A similar alignment of seismicity was previously reported by Fauzi et al.[13], who performed an analysis of earthquakes recorded by local and regional seismic networks. These authors hypothesized that the IFZ serves as a site of focused volatile release into the overlying mantle wedge.

### Regional tomography study

Figure 3c shows P-velocity anomalies in the upper mantle derived as a result of regional tomography studies. An earlier version of this model has previously been presented[14]; however, we subsequently modified it slightly by tuning the parameters and incorporating new data. This tomography model was constructed using seismic travel time data from the global catalogue of the International Seismological Centre (ISC) and the algorithm developed by Koulakov and Sobolev[15]. Further details on the data and algorithm are provided in the Methods section. Consistent mantle heterogeneities were reported in another tomography study[16] based on similar data and approach. In the model presented in Fig. 3c, we observe a linear high-velocity anomaly representing the subducting lithosphere. We observe that in the area around the IFZ prolongation, the slab has a curved shape, which may indicate strong deformation and tearing.

### Local tomography studies

On the local scale, several seismological studies in the Toba region and surrounding areas have previously been performed. The interaction between the IFZ and the overriding plate was investigated based on the offshore seismic network[17]. The deep structure beneath Toba was studied[18,19] based on data recorded by seismic stations deployed in 1995. Subsequently, a new seismic network was installed in the Toba area in 2008 (ref. 20). Using data from this network, ambient noise tomography studies[20,21] imaged the detailed structure of the upper crust beneath Toba. In addition, the latest study[21] revealed strong radial anisotropy at depths >7 km, thus providing evidence of a large sill complex beneath the caldera. Receiver function studies[22] have identified increased crustal thickness, up to 35–38 km, beneath Toba.

Here we present an additional local-scale tomography model, which was calculated based on the combination of seismic travel time data obtained from the two above-mentioned temporary seismic networks in the Toba area. The first data set was recorded by a seismic network deployed around Toba during the period from January to May 1995. The network consisted of 10 broadband and 30 short-period seismic stations (white triangles in Fig. 1). The second data set was recorded by another

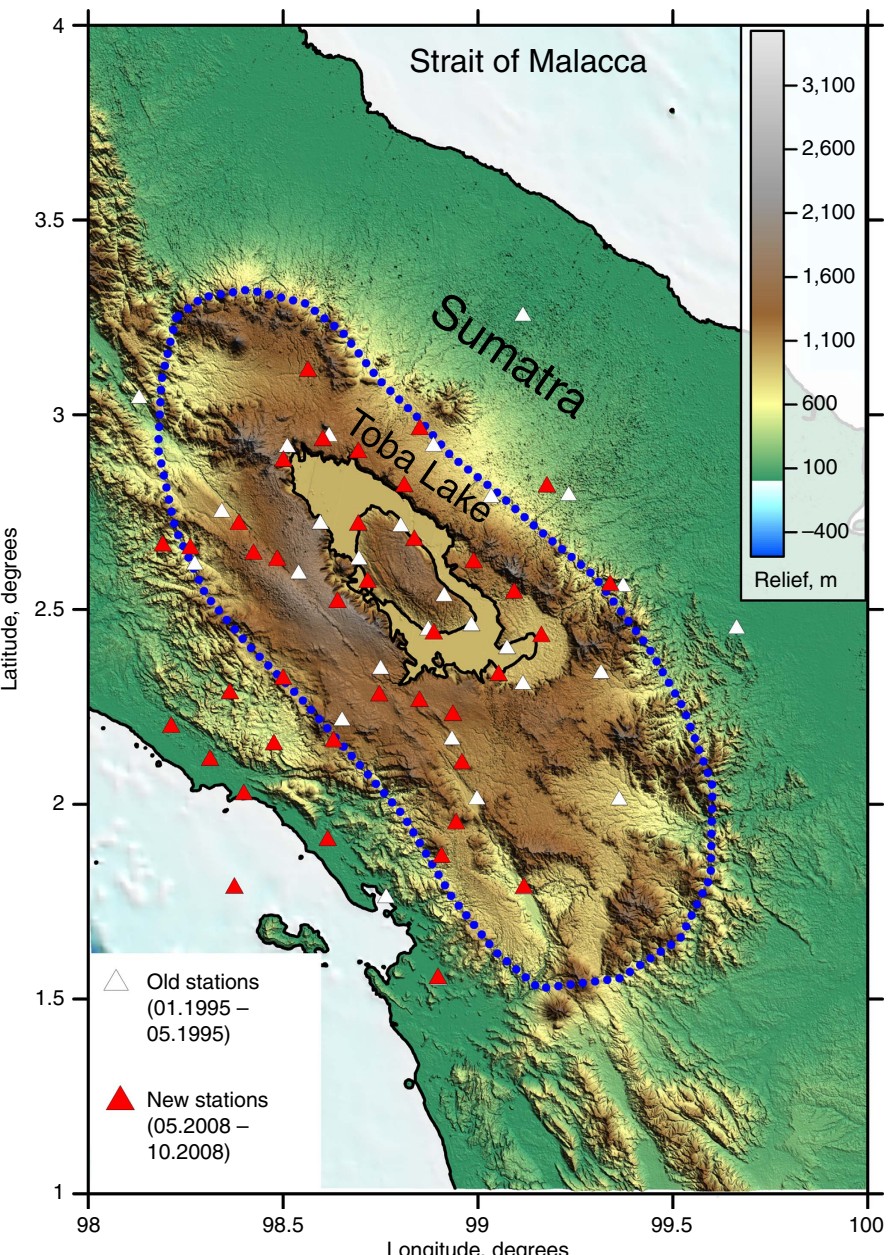

**Figure 1 | Topography of the Toba Caldera area.** White and red triangles indicate the seismic networks that were operated in 1995 and 2008, respectively. The blue dotted line indicates the uplift area.

seismic network deployed in the same approximate area by GeoForschungsZentrum-Potsdam for 5 months from May to October 2008. This network consisted of 42 short-period three-component seismometers (red triangles in Fig. 1). For the second data set, we manually identified the local events and picked the arrival times using SEISAN software[23]. To select the data for tomography, for both data sets we used only events that had a number of P- and S-picks per event equal to or larger than 8. We rejected the picks with absolute values of the residuals larger than 1 and 1.5 s for the P- and S-data, respectively. From the first data set we selected 505 events with 7,058 corresponding arrival times (4,122 P- and 2,936 S-waves) and for the second data set we obtained 4,826 arrival times (2,522 P- and 2,304 S-waves) from 149 local events. Although the number of events in the second case was smaller relative to the older data set, the quality of these data was considerably higher. The average number of picks per event in this case was ∼32, whereas for the older data set it was

<14. The distributions of seismic stations and events are plotted in Supplementary Fig. 1.

The tomography inversion was performed using the LOTOS code[24]. The velocity models were parameterized using nodes distributed within the study volume according to the density of rays (Supplementary Fig. 2). In map view, in areas with sufficient coverage these nodes are regularly distributed with a spacing of 10 km. In the vertical direction, the node spacing is inversely dependent on the distribution of data, but cannot be <5 km. Further details on the algorithms are described in the Methods section.

As a result of the tomographic inversion, we obtained the three-dimensional (3D) distributions of the P- and S-wave velocity anomalies with respect to the preferred one-dimensional (1D) velocity models and the updated locations of seismic events. For the main discussion, we selected the distributions of S-wave velocities (Fig. 4), because they are more sensitive to temperature-

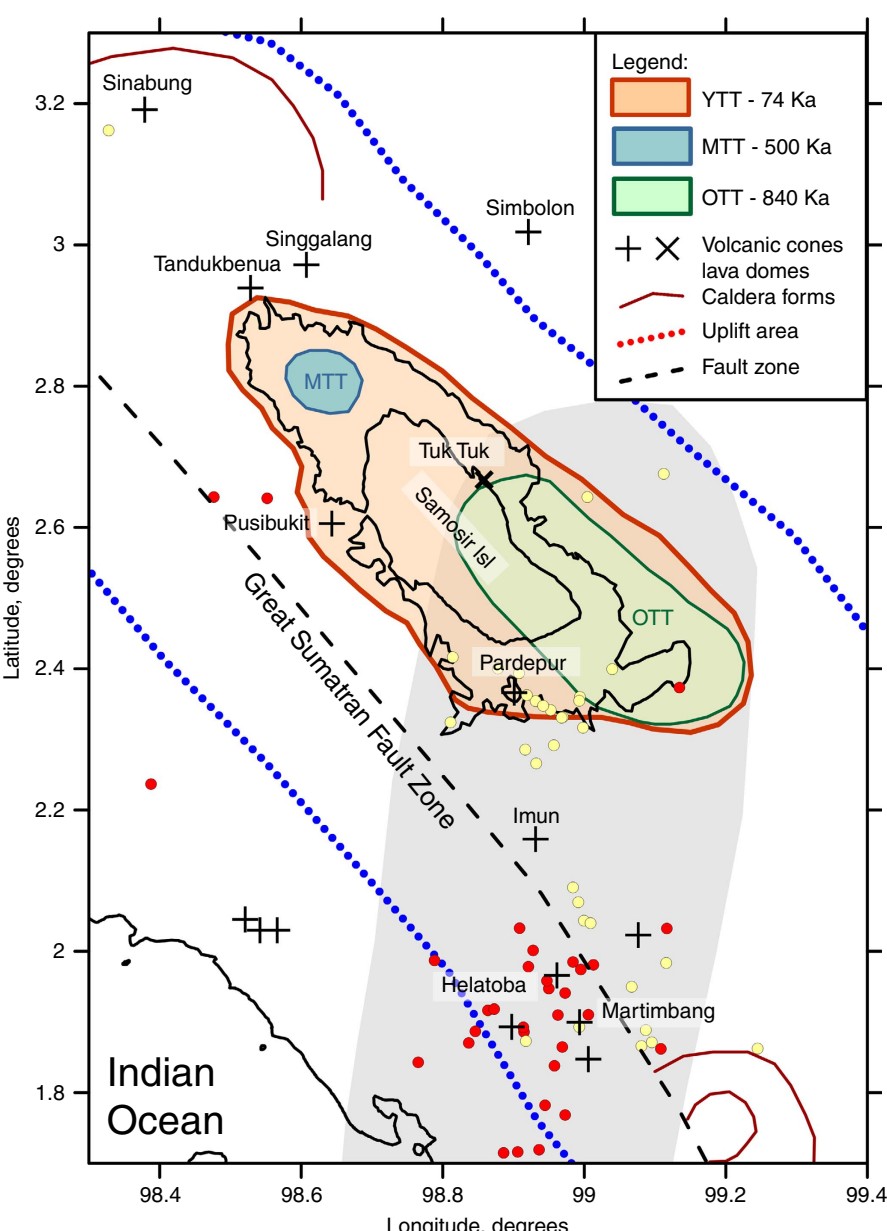

**Figure 2 | Locations of volcano-related structures and seismic stations in the Toba region.** Volcanic cones (crosses) were identified from the topographic map and from geological information[2]. Contours of three major calderas[2] are indicated by the Young, Middle and Old Toba Tuffs (YTT, MTT and OTT, respectively). Other caldera-related structures are highlighted with brown contours. Dots depict slab-related seismicity (red: 50–140 km depth; yellow: 140–170 km depth). The grey area highlights the area of deep seismicity presumably associated with the IFZ.

and fluid-related heterogeneities and, therefore, to the volcano-related features. The P-wave anomalies in the horizontal sections are shown in Supplementary Fig. 3. In Fig. 5, we also present the P- and S-velocity anomalies and Vp/Vs ratio in a vertical section oriented parallel to the displacement direction of the subducting plate. The absolute P- and S-wave velocities are shown for the same section in Supplementary Fig. 4.

At shallow depths corresponding to upper crustal structures (10 km depth), the low-velocity S-wave anomalies (Fig. 4) coincide with the locations of the Toba Caldera, other volcanic complexes (yellow crosses) and the GSFZ. The prominent low-velocity anomaly '4' beneath the Toba Caldera appears to be consistent with previous studies that were based on body wave[18,19] and ambient noise[20,21] data. At depths of 30 and 50 km, we identify an elongated low S-velocity anomaly '3' with dimensions of ~120 × 20 km that probably represents magma

storage at the base of the crust. In the vertical section in Fig. 5, we observe that these two anomalies (patterns '3' and '4') are characterized by low S-wave velocity patterns and high Vp/Vs ratio. The deeper anomaly '3' coincides with a seismicity cluster located at the base of the crust. It is noteworthy that a very similar structure was identified beneath the Klyuchevskoy volcano group in Kamchatka, where a Vp/Vs anomaly as high as 2.2 is located at the base of the crust and coincides with an extremely strong seismicity cluster[25]. Similarly, as in the Toba case, a low Vs and high Vp/Vs anomaly was detected in the middle crust beneath the Klyuchevskoy volcano and interpreted as an intermediate magma reservoir. The synthetic modelling, which will be discussed below, shows that using the higher-quality data in this study provides the possibility to distinguish the patterns '3' and '4', whereas in previous studies they were seen as a single vertically smeared anomaly. Another crustal anomaly, similarly characterized by low

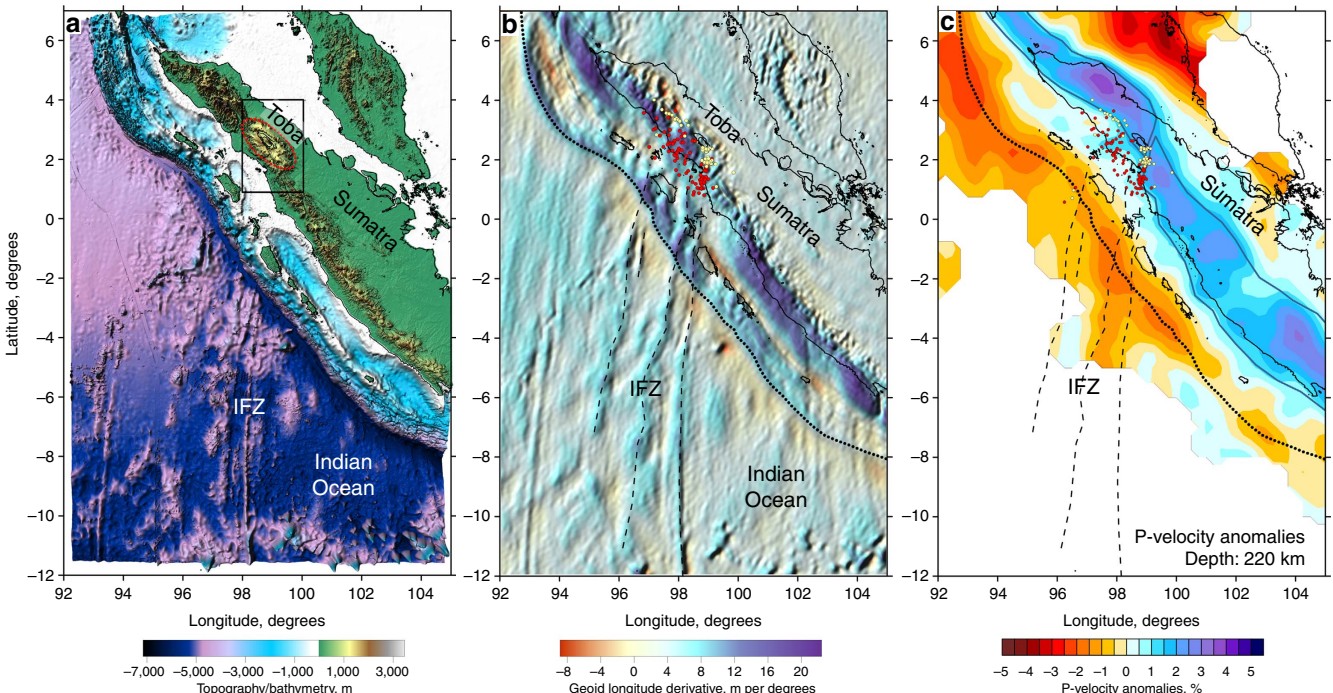

**Figure 3 | Location of the Investigator Fracture Zone with respect to the Toba Caldera.** (**a**) Bathymetric and topographic map of the area adjacent to the Sunda subduction zone. The uplift around the Toba Caldera is highlighted by the red dotted line. The rectangle indicates the study area. (**b**) Longitude-directed derivative of the geoid model EIGEN-6C4 (ref. 11). Dots indicate the slab-related seismicity used in this study at two depth intervals (red: 50–120 km; yellow: 120–170 km). (**c**) P-velocity anomalies at 220 km depth according to the regional tomography model[20]. The dark blue line highlights the possible boundaries of the slab. In **b** and **c**, the black dotted line indicates the trench. Dashed lines depict ridges associated with the Investigator Fracture Zone (IFZ).

S-velocity and high Vp/Vs ratio pattern, is clearly associated with the GSFZ and indicates fracturing of crustal rocks in the fault zone.

At greater depths (80 km section in Fig. 4), we observe two clearly separated linear low-velocity anomalies that can also be observed in the vertical profiles of both the P- and S-models (Fig. 5). One of these anomalies (denoted by '1') represents a vertically oriented 'wall' that originates in the slab area at a depth of ~150 km. A similar feature has been identified in a previous tomography study[19]. The second low-velocity anomaly, denoted by '2', connects the slab area at ~80 km depth with the forearc. This anomaly has not been revealed in previous tomographic studies because of insufficient data coverage.

Several synthetic tests provided in Supplementary Figs 5–8 show that incorporating new data in the model has enabled better coverage compared with that of previous work[19], in particular for areas between the Toba Caldera and the coast. The checkerboard test in Supplementary Fig. 5 shows that at shallow depths the anomalies >30 km in size are robustly resolved, whereas in deeper sections the minimum size of the resolved anomalies is ~50 km. Tests using realistic anomalies, shown in Supplementary Figs 6 and 8, help to assess the leakage of anomalies due to smearing and limited spatial resolution. We defined the model patterns to achieve the best resemblance between the recovered model and the results derived from inversion of the experimental data. Thus, the amplitude values defined in the realistic synthetic model represent the true values expected for anomalies in the Earth's subsurface. It is important that we can now clearly distinguish relatively complex structures in the crust, as shown in the synthetic test in Supplementary Fig. 8, which illustrates considerable improvement in the model resolution following the addition of new data. However, for the mantle, the synthetic test with vertical checkerboards (Supplementary Fig. 7) demonstrates the limited vertical resolution caused by the trade-off between source and velocity parameters.

It is a general problem of tomography studies that the values of anomalies depend on the data coverage and damping parameters. Synthetic tests using realistic anomalies allow estimation of the leakage of the anomaly values due to smearing and damping, and assessment of their original amplitudes. For example, for anomalies '3' and '4', we estimated the S-wave velocity anomaly values at 16% and 18%, respectively. However, for some parts of the model, we cannot guarantee that the reported values actually do represent the exact seismic parameters in the Earth. Therefore, we should exercise caution when attempting direct conversion of seismic anomalies into petrophysical parameters, such as temperature or melt content. In our interpretation, we generally qualitatively consider the shapes and relative strengths of anomalies without emphasis on their numerical values.

## Discussion

In the introduction, we identified several questions related to supervolcanism, which remain open and are actively discussed in the scientific community. The first key question is why the particularly large explosive eruptions in the Toba Caldera occur repeatedly in approximately the same location. We propose that the main cause of such behaviour is associated with the IFZ, which is being subducted directly beneath Toba. The IFZ is a prominent structure with an elevation of 1,000–1,500 m above the surrounding sea floor. Owing to isostasy, the expected crustal thickness in the IFZ may be double that of the normal oceanic crust. Furthermore, fractures within the IFZ facilitate penetration of seawater into the crust[26]. As a result, the subducting IFZ transports an anomalous amount of crustal material and fluids to

the mantle. Therefore, it behaves differently to the normal slab and dehydration during subduction produces a much larger amount of volatiles[26].

Some of the volatiles may escape from the slab during the first stages of dehydration phase transitions at relatively shallow depths. In particular, the negative anomaly '2' (Fig. 5a,b), which

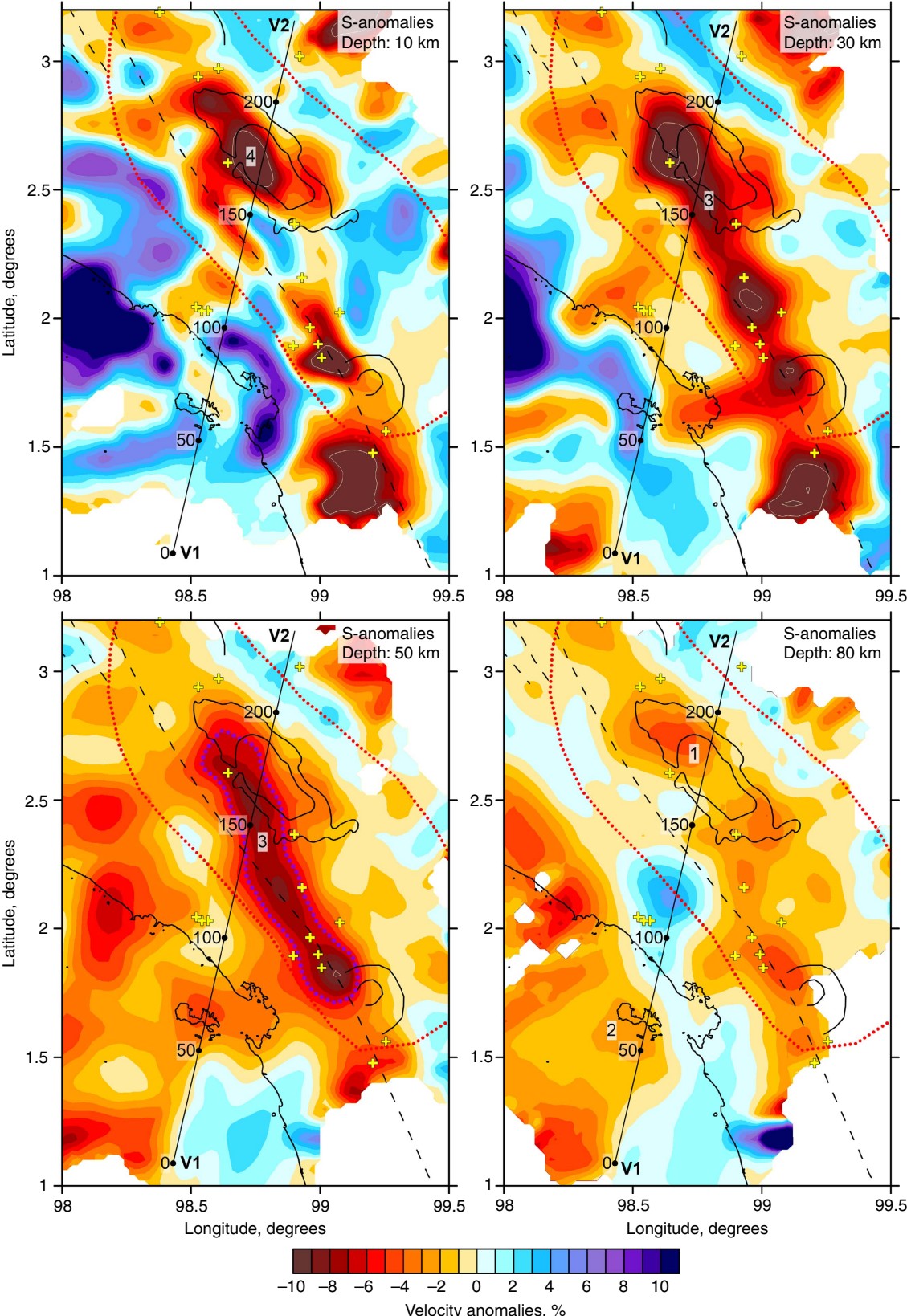

**Figure 4 | S-wave velocity anomalies in four depth sections.** The red dotted line indicates the uplift area. Crosses indicate the volcanic cones and lines indicate caldera-related structures. The dashed line depicts the GSFZ. Profile V1–V2 indicates the location of the vertical section presented in Fig. 5.

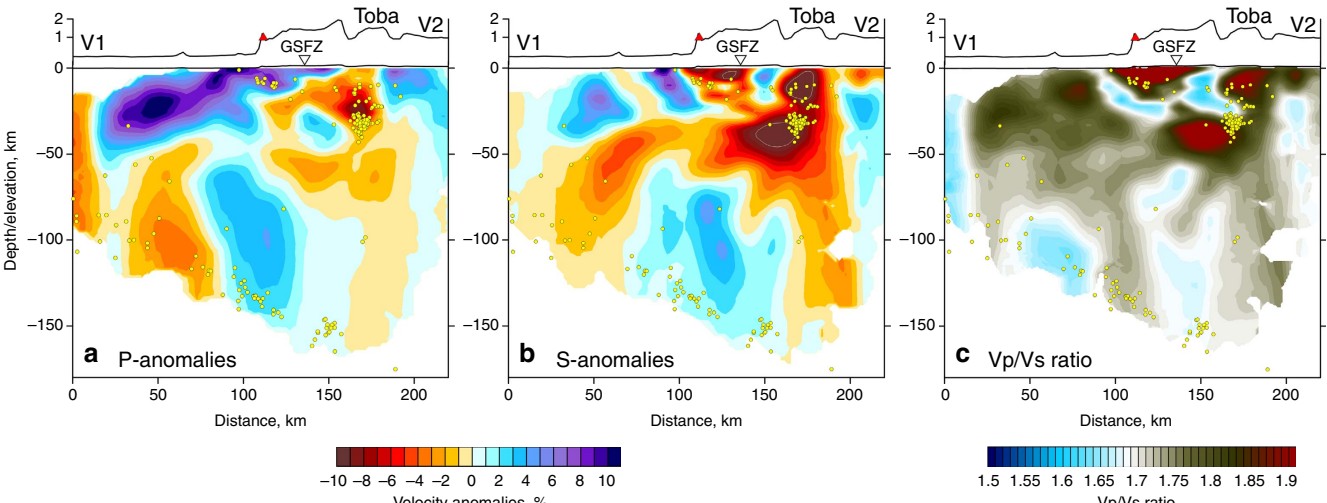

**Figure 5 | Distributions of seismic parameters in vertical section V1–V2.** P- and S-wave velocity anomalies and Vp/Vs ratio are presented in **a**, **b** and **c**, respectively. Location of the profile is indicated in Fig. 4. Yellow dots depict seismic events located at distances of <20 km from the profile. The exaggerated relief is presented above each plot. The inverted triangle labelled GSFZ is the intersection with the GSFZ. The red triangle indicates a volcanic complex identified in the coastal area.

originates from the slab at 80 km depth beneath the forearc, may be a signature of serpentinization and/or high fluid release from the slab. High Vp/Vs ratio (Fig. 5c) above this anomaly may indicate fluid saturation in the uppermost mantle and crust beneath the forearc. The temperature of the pathway in this area is not high enough to melt rocks and generate considerable volcanic activity. Such 'early' escape of fluids was similarly observed in many other subduction zones, such as in the northern[27] and southern Andes[28].

In addition to bringing an anomalous amount of fluids to the mantle, subduction of the IFZ could trigger slab tear, which may also intensify the process of magma generation[29]. Based on the analysis of deep earthquakes beneath Sumatra, Fauzi et al.[13] pointed out that the IFZ may correspond to a tear or a step in the slab. Strain along this step is likely to be the cause of the linear zone of seismicity. The same conclusion is drawn from the regional tomography result shown in Fig. 3c. The shape of the slab-related high-velocity anomaly shows curvature of the slab in the area of the IFZ, which may indicate strong deformation and tearing, as highlighted by the blue lines.

We suggest that the difference in oceanic plate age across the IFZ could be responsible for the step or tear. The younger crust to the northwest of the IFZ was formed at a fast-spreading mid-ocean ridge, whereas the older crust to the southeast was formed at a slow-spreading ridge (see the plate model[9]). This is likely to result in significantly different crustal thickness and thermal structure, and therefore in a difference in slab buoyancy. According to this interpretation, the southeastern segment of the slab is more dense and therefore it subducts more steeply than the northwestern segment. If a slab tear is present, this could make it easier for fluids to enter or exit the deeper part of the slab and could also cause significantly faster heating of the slab. In turn, this may be one of the key factors contributing to the origin of the Toba supervolcanism. Penetrating hotter asthenospheric material may be another factor facilitating the voluminous melting in the mantle wedge. In Fig. 6a, we schematically illustrate the relationship between slab tear and the magma system beneath the Toba Caldera.

The second key question is why the extremely violent eruptions in the Toba Caldera alternate with long periods of relative quiescence. What prevents the continuous occurrence of moderate volcanism? The local seismic tomography model

indicates the configuration of the multilevel magma plumbing system beneath Toba and can help to answer this question. The functioning of the Toba magmatic system deduced from the seismic tomographic model is illustrated in Fig. 6b. The subducting lithosphere along the IFZ line is characterized by thicker crust and is more hydrated than 'normal' oceanic lithosphere[26]. Furthermore, it may be additionally heated from a slab window that originated due to slab tear. Both of these factors may lead to the anomalous release of volatiles at a depth of ~150 km, which is expressed by high seismicity along the slab. When the fluids penetrate through the mantle wedge, they may react with peridotites and transform them into phlogopite- or amphibole-bearing rocks, which have lower melting temperatures[29]. These processes lead to the growth of ascending diapirs containing high-temperature volatile-rich basic melts produced by partial melting of the mantle wedge. The ascending partially molten magma pathways are expressed by the low-velocity anomaly '1' in Fig. 6b. At depths of 30–50 km, these ascending magma diapirs form a large reservoir (anomaly '3'). The strong negative shear velocity anomaly and much weaker P-wave velocity anomaly suggest the presence of significant amounts of partial melt and volatiles inside this reservoir. However, as the seismic S-wave can propagate through the reservoir, the solid component is still dominant. It is difficult to quantify the volume of the liquid phase because of uncertainty in the exact determination of seismic anomaly values and due to ambiguous transitions between seismic and petrophysical properties. We selected a value of −7% of the S-wave anomaly at 50 km depth as the most plausible threshold representing the rocks with high fluid content. Following the contour line for this value (violet dotted line in Fig. 4), we estimated the volume of reservoir '3' at 50,000 km³. A similar basic magma reservoir has recently been identified at the base of the crust beneath the Yellowstone supervolcano[30], which may indicate that this is a common feature of supervolcanic structures.

The basic magma in reservoir '3' is too dense to continue ascending through the lower-density continental crust. Geochemical evidence based on 87Sr/86Sr ratios indicates that the typical Sumatran stratovolcanoes comprise mafic melts (and their derivatives) generated from the mantle wedge, whereas Toba consists almost exclusively of melts derived from continental crust[2]. From this, we can conclude that there is no considerable

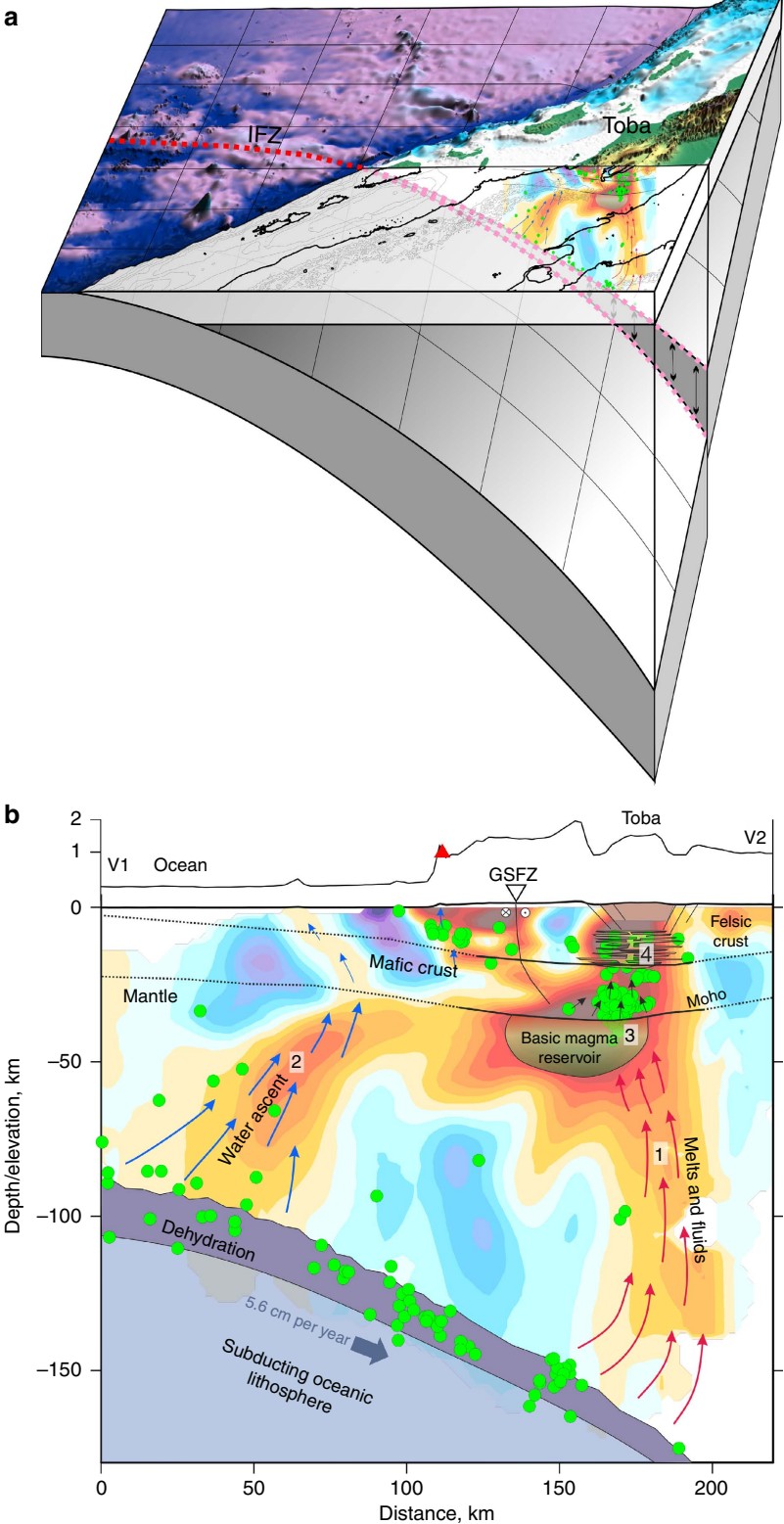

**Figure 6 | Schematic representation of the magmatic plumbing system beneath Toba and its relation to the IFZ.** (**a**) Three-dimensional view of the oblique subduction of the IFZ (red dotted line) beneath Toba. S-velocity anomalies (same as in Fig. 5) are shown in the mantle wedge beneath Toba. The red dotted line depicts the IFZ and the zone of slab tear. (**b**) Interpretation of the S-wave velocity model in the vertical section shown in Fig. 5. Exaggerated topography is shown above the section. The red triangle indicates the location of the volcanic complex identified in the coastal area. Green dots depict the earthquakes. Blue arrows indicate possible ascent of water from the slab dehydration zone at 80–90 km depth. Red arrows indicate the path of ascending fluids and melts that originated from the slab at ~150 km depth. The crustal interfaces beneath Toba (solid lines) are shown based on receiver function results[19]. Surrounding the Toba Caldera, their shapes are extrapolated according to general knowledge about the crustal nature and isostasy (dotted lines). The location of the GSFZ with the displacement polarity and possible depth propagation is indicated.

contribution of melts from the mantle in forming the upper crust reservoir; however, the reservoir may serve as a powerful heat source and may transfer upward a significant amount of volatiles. The mobile ascending volatiles appear to be a very efficient mechanism of heat transport. The observed seismicity in the lower crust above anomaly '3' possibly indicates the pathways of the ascending volatiles, which provides evidence for the active state of the magma factory at the present day. Reaching the middle and upper crust, the overheated volatiles cause melting of rocks that form the shallow silicic magma reservoir observed as anomaly '4'. When a critical amount of molten upper crust material highly saturated with fluids is accumulated, an avalanche-type process may start. Ascending quantities of such material results in decompression and transformation of overheated fluids to gases. In turn, this increases pressure and accelerates fluid and magma ascent. Finally, this avalanche-type process triggers a large explosion causing a supereruption. The accumulation of a sufficient amount of 'explosives' in the upper crust takes hundreds of thousands of years, which may give the answer to the second question related to long periods of quiescence between supereruptions. A similar scenario was proposed by Shapiro and Koulakov[31] for the case of Yellowstone, based on the tomography results of Huang et al.[30] who identified a multilevel plumbing system including a large reservoir at the base of the crust, which appears to be very similar to the results of this study for Toba.

The final question, related to the present-day activity of the magma reservoirs beneath Toba and the possibility of a future supereruption, remains open. It is probable that, in the long term, large eruptions will occur repeatedly until the IFZ, which is the major source of the supervolcanism, subducts beneath Toba. Regarding the present-day state, the intensive seismicity within the reservoir at the base of the crust indicates its current activity. On the other hand, the time that has currently passed since the last supereruption (74 Ka) is too short in comparison with the periodicity of large volcanic events at Toba. The critical mass of the molten magmas and volatiles in the upper crust has most likely not yet been achieved and the next supereruption may be expected only in some dozens of thousands or hundreds of thousands of years.

We conclude that the exceptionally voluminous explosive volcanism of Toba results from a combination of several factors. First, anomalously large amounts of volatiles are generated at depth, caused by the subduction of the IFZ. These volatiles cause active melting in the mantle wedge, thereby forming ascending magma diapirs. The direct ascent of the magmas to the surface is not efficient because of the presence of a $\sim$38-km-thick continental crust[22], which appears to be an obstacle for basic magmas that are not sufficiently buoyant. Therefore, they are stored in a large magma reservoir at the base of the crust. This reservoir serves as a powerful heat source and a source of upward-migrating overheated volatiles. Reaching the middle and upper crust, these volatiles cause active melting of silicic rocks. These melts form a shallow reservoir of magmas that are viscous and strongly enriched in volatiles and, therefore, are explosive. This reservoir does not produce small frequent eruptions and is spasmodically emptied during recurrent but sporadic catastrophic eruptions that are followed by long periods of quiescence. The results of our tomographic model presented in this study suggest that the Toba magma-generating engine continues to be active at present and, despite its current period of inactivity, this volcano system may generate strong eruptions in the future.

## Methods

**Regional tomography model.** In Fig. 3c, we present a horizontal section of the P-velocity anomalies at 220 km depth. This result is a detailed section from a regional tomography model for the entire Sunda Arc, which is an updated version of a previously published model[14]. The P- and S-wave velocity heterogeneities down to 1,000 km depth were computed based on the tomographic inversion of P- and S-body wave travel time data from the ISC[32] for the time period from 1964 to 2012. Before implementing the tomography, all the ISC data were reprocessed. The events were relocated using an algorithm[15] in which travel time data are corrected for relief and variations in crustal thickness according to the CRUST2.0 model[33]. This algorithm performs outlier analysis, which results in rejection of $\sim$30% of the data.

For the regional tomography, we use all available data with ray paths propagating, at least partly, through the study volume. This includes the data from worldwide seismicity recorded by stations located in the study region, as well as data from events in the study region recorded by worldwide stations. The inversions are performed independently in a series of overlapping circular windows and subsequently combined into one model. Such an approach allows the identification of finer structures than in the case of global inversion, but it risks losing larger-scale anomalies with sizes comparable to the window diameter[34]. This algorithm has been thoroughly tested in many different regions[34] and, in particular, it appeared to be efficient for studying several subduction zones[35]. For the case shown here, the inversion was performed for the entire Sunda Arc in a volume down to 1,000 km depth. The anomalies are shown only in areas with sufficient ray coverage.

This model is slightly different to that presented by Luhr et al.[14], because it incorporates additional data corresponding to the 5 years from 2008 to 2012. In addition, we tuned the values of the inversion parameters to better reveal the slab-related anomaly. The derived result appears to be consistent with another model obtained independently[16].

**Data for the local tomography inversion.** In this study, we used the arrival times of P- and S-seismic body waves from earthquakes located inside or slightly outside the study area. Here we combine two data sets corresponding to two seismic networks that were operated in the Toba Caldera region during different time periods. The older data set was recorded by a seismic network deployed around Toba by the Incorporated Research Institute for Seismology for the Program for Array Seismic Studies and the Indonesian Seismological Agency. The network, consisting of 10 broadband and 30 short-period seismic stations, was operated during the period from January to May 1995. During the observation period, 1,500 local events were recorded[13]. For this study, we used only events that had a number of P- and S-picks per event $\geq$8. We rejected the picks with absolute values of the residuals $>1$ and 1.5 s for the P- and S-data, respectively, estimated after source locations in the 1D starting model. These selection criteria reduced the number of events to 505, which had 7,058 corresponding arrival times consisting of 4,122 P- and 2,936 S-waves. An approximately similar data set was used in the previous tomography studies[18,19].

The second data set was recorded by another seismic network deployed in approximately the same area by GeoForschungsZentrum-Potsdam for 5 months from May to October 2008. The network consisted of 42 short-period three-component seismometers and L4-3D stations with natural frequency of 1 Hz. We downloaded the full continuous data from this experiment through the GEOFON system of GFZ. We manually identified the local events and picked the arrival times using SEISAN software[23]. To select the data for tomography, we used the same criteria applied to the older data set (8 picks per event and 1–1.5 s of residual threshold). In total, we obtained 4,826 arrival times (2,522 P- and 2,304 S-waves) from 149 local events. Although the number of events in this case was smaller relative to the older data set, the quality of these data was considerably higher. In particular, the average number of picks per event in this case was $\sim$32, whereas for the older data set it was $<$14. In Supplementary Fig. 1 we present separate distributions of the shallow and deep events used in this study. The ray paths of the S-waves are plotted in map view and in a vertical section in Supplementary Fig. 2. We observe some clustering of events causing uneven data distribution; however, in summary, the ray coverage is relatively good for performing high-quality tomographic inversion (as will be shown by synthetic tests).

Another potential concern is related to the use of events located outside the network for which we cannot ensure the high quality of coordinate and origin time determinations. Based on previous studies, we suggest that data from such events appears to be important for improving the resolution of the model[36]. Similar to teleseismic studies, such events may provide relative time residuals in stations of the network that can be used for tomography. In addition, the use of such data considerably enhances the variety of ray propagation directions and allows for greater depths to be reached compared with the use of events that only occur inside the network. All of these factors are favourable for improving the quality of tomographic inversions.

**Local earthquake tomography algorithm.** For the tomographic inversion, we used the LOTOS code[24]. The main input data for the inversion contained station coordinates and arrival times of P- and S-waves. In addition, the initial 1D velocity model and the parameters for source locations and inversion should be defined.

During the preliminary stage, we located the sources in the initial 1D velocity model using a grid-search method. To speed up the calculations in this stage, the

travel times were calculated along straight ray paths. In this stage, we rejected data according to the criteria described above.

The subsequent calculations involve iterative repetitions of source relocations and inversion steps. The relocation of sources in the 3D velocity model is based on the gradient method. Travel times are calculated using a 3D ray tracer based on the bending scheme[37].

The velocity models were parameterized using nodes distributed within the study volume according to the density of rays. In map view, in areas with sufficient coverage (0.1 of the average ray density) these nodes are regularly distributed with a spacing of 10 km. In the vertical direction, the spacing is inversely dependent on the distribution of data but cannot be <5 km. The distributions of nodes in map view and in a vertical section are presented in Supplementary Fig. 2. To reduce any grid dependency, we performed the inversion for several differently oriented grids (with the basic orientations of 0°, 22°, 45° and 67°) and then averaged the results.

The inversion was performed simultaneously for P- and S-velocities, and for source parameters (four parameters for each event) using the LSQR algorithm[38]. In this case, we regularized the solution by adding a smoothing matrix block linking all neighbouring nodes that minimize the gradients in the calculated velocity models. This corresponds to adding equations for all combinations of node pairs:

$$W^{sm}(x_k - x_m) = 0 \qquad (1)$$

where $W^{sm}$ is a smoothing coefficient, and $x_k$ and $x_m$ are the velocity anomalies in neighboring nodes $k$ and $m$. For the final solution, we used five iterations as a compromise between the calculation time and the solution accuracy.

In this study, we used the same starting velocity model as that determined by a 1D optimization algorithm for the same region in a previous tomographic study[19]. In this model, a constant Vp/Vs value and P-values for several depth levels were defined as presented in Supplementary Table 1. Between these levels, the velocity values were linearly interpolated.

All of the results presented in this study can be easily reproduced using a code template, which includes all of the programmes, data, parameters and simple instructions for performing the calculation. The full directory can be downloaded from www.ivan-art.com/temp/lotos_toba.zip.

**Tomography inversion results.** The main results were calculated after five iterations. The values of the average deviations of the residuals and their reduction during the iterative procedure are presented in Supplementary Table 2. It is noteworthy that the variance reduction for the S-model was stronger than that for the P-model. Although the noise in the S data was greater, which can be observed in the final values of the average residuals, a stronger reduction was caused by the considerably larger amplitudes of anomalies in the resulting S-model and the higher sensitivity of the S-wave travel times to the velocity heterogeneities.

The tomographic inversion provides simultaneous calculations of P- and S-velocity heterogeneities and source parameters. The source locations in the final 3D model after five iterations are presented in Supplementary Fig. 1. Here we separated the crust-related earthquakes (<40 km depth) from the slab-related seismicity with depths >40 km. The majority of the shallow seismicity (Supplementary Fig. 1A) is associated with the GSFZ, with the major cluster aligned along the southern segment of the GSFZ. A striking feature is a fault-perpendicular seismicity cluster located beneath Toba at a depth interval of 20–40 km. As will be shown, this cluster is associated with a strong seismic anomaly and, therefore, is considered to be a manifestation of magmatic processes beneath Toba.

For the deep seismicity in Supplementary Fig. 1B, we observe a gradual increase in focal depth towards the northeast from 40 to 150 km depth, which is typical of a Benioff zone. A striking feature is an elongated zone characterized by a high concentration of seismicity, highlighted in grey colour. This feature perfectly corresponds to the prolongation of the IFZ and terminates just beneath the southern part of the Toba Caldera. This structure was previously identified from an analysis of earthquakes recorded by the local and regional seismic networks[13]. These authors hypothesized that the IFZ 'serves as a site of focused volatile release into the overlying mantle wedge'. Our tomography results appear to support this notion.

The resulting velocity models are shown as anomalies with respect to the reference model provided in Supplementary Table 1. S-wave velocity anomalies in four horizontal sections are presented in Fig. 4 and the P-anomalies are shown in Supplementary Fig. 3. In the vertical section, we plot anomalies of the P- and S-velocities and Vp/Vs ratio (Fig. 5). The absolute values of the P- and S-velocities are shown in Supplementary Fig. 4.

At a depth of 10 km, we observe a linear low-velocity anomaly in both the P-model and the S-model, associated with the GSFZ and major volcanic manifestations; however, several differences can be observed between the P-model and the S-model. In the S-model the most prominent low-velocity anomaly is associated with the Toba Caldera, whereas in the P-model the negative anomalies are also present but weaker. In both models, prominent low-velocity anomalies are associated with the Helatoba volcanic complex located to the south of the Toba Caldera. The GSFZ is better revealed by the S-velocity anomalies. In the P-model the forearc is fully 'blue,' whereas in the S-model it appears in patches. These patterns of lower S-velocity may represent traces of melt and volatile release in the forearc. In the P-velocity model, we identify a local high-velocity pattern centred

inside the caldera-shaped structure to the south of Toba, which may represent an exhumed core of the magma reservoir that was responsible for forming the caldera.

At a depth of 30 km, the S-model reveals an elongated low-velocity structure underlying all the volcanic centres around Toba. The amplitudes of the S-anomalies at this depth are the highest and reach 15% beneath Toba. In the P-velocity model, a low-velocity pattern is also observed in the same approximate location; however, it is weaker in amplitude and interrupted by several local high-velocity patterns. This difference can be explained by the higher sensitivity of the P-velocity to composition, whereas the S-velocity varies predominantly because of the presence of liquid phases. Therefore, we can propose that at this depth a high content of volatiles and melts is expected because of the good fit of the low S-velocity patterns with the distribution of the recent volcanic cones.

At a depth of 50 km, the elongated low S-velocity anomaly is still observed beneath Toba and the Helatoba volcanoes to the south. The same anomaly is observed in the P-model; however, it appears to be much weaker. The area beneath the forearc appears to be associated with low-velocity anomalies in both the P-model and the S-model. At a depth of 80 km, both the P-model and the S-model show two clearly separated low-velocity anomalies: one associated with the arc volcanoes and the other associated with the offshore areas.

The vertical section A–B in Fig. 5 is oriented parallel to the direction of subduction and helps reveal the structures associated with the sinking IFZ. The P- and S-velocity structures are consistent in this section; however, the S-velocity model more clearly demonstrates the processes beneath Toba and the surrounding areas. Therefore, we will mostly focus our interpretation on the S-model. In this section, the seismicity is shown to be uniformly aligned in the Benioff zone. As shown in Supplementary Fig. 1B, this seismicity corresponds to a narrow belt that is presumably related to the subduction of the IFZ and caused by strain and post-subduction faulting of the slab. The shallow structures in the forearc area are associated with a higher velocity anomaly down to a depth of ∼50 km. In the S-model, this forearc anomaly is interrupted by local positive anomalies associated with Mursala Island and the GSFZ. Beneath Toba, we observe a low-velocity pattern, which is particularly intensive in the S-velocity model. This anomaly consists of two parts and the lower part is associated with a seismicity cluster identified at depths of 30–40 km.

At greater depths, both the P- and S-velocity models reveal two low-velocity anomalies connecting the slab with the Toba Caldera. One anomaly originates at a depth of ∼150 km and then ascends almost vertically towards the Toba Caldera, whereas the other anomaly has an inclined shape and connects the forearc area with the slab at a depth of ∼80 km. An interpretation of these anomalies, which is predominantly based on the S-model, is provided in the Discussion section.

**Synthetic tests.** Synthetic tests were designed to verify the horizontal and vertical resolution and assess the optimal values of the inversion parameters. In the synthetic modelling, we simulated the same workflow applied in the observed data analysis. After calculating the synthetic travel times using the 3D bending ray tracer, we shifted the coordinates and origin times of the sources so that they remained unknown during the reconstruction. In all of the tests presented below, the travel times were additionally perturbed with random noise of 0.1 s of average deviation. The synthetic data were processed using the same steps and the same inversion parameters as applied to the observed data, including the initial source location step in the 1D starting model that strongly biased the synthetic signal in the data and created the trade-off between the source and velocity parameters.

The first group of checkerboard tests shown in Supplementary Fig. 5 is oriented for studying the horizontal resolution. The synthetic anomalies are defined as periodic alternating patterns that remain unchanged with depth. In Supplementary Fig. 5, we present two tests with anomaly dimensions of 30 × 30 km and 50 × 50 km. The amplitudes of the anomalies in both cases were ±5%. For the P-model and the S-model, we assigned opposite signs to the anomalies. The obtained results show that the resolution depends on the depth. At a depth of 10 km the 30-km anomalies are robustly resolved, whereas at a depth of 80 km the anomalies are mostly smeared. The 50-km anomalies are correctly reconstructed for both sections. The resolutions of the P-model and the S-model are similar.

The capacity to resolve anomalies with realistic shapes is explored in another synthetic test presented in Supplementary Fig. 6. The synthetic model was defined as a set of 3D prisms with polygonal shapes in map view and preset depth intervals. We designed the shapes and amplitudes of the anomalies to achieve recovery results similar to that of the main S-velocity model shown in Supplementary Fig. 3. Furthermore, in this model, we attempted to simulate several geological structures, such as the GSFZ, the Toba Caldera and local features associated with recent volcanic complexes. This method was previously used[16] and it can be used to assess the bias of anomaly shapes and amplitudes due to damping and smearing effects. In turn, the method can be used to retrieve realistic amplitudes of anomalies in the experimental data processing results. As shown in this test, the recovered anomalies were similar to those obtained from inversions of actual data; therefore, the preset values of the anomalies were close to those expected for real structures beneath the Toba region.

The aim of the second group of tests was to verify the vertical resolution along the profile used to present the main results. In tomographic studies with uncontrolled sources, poorer vertical resolution is a fundamental problem related to the trade-off between velocity and source parameters. Certain authors have

frequently obtained overly optimistic estimates for the vertical resolution, because they presume known locations of sources and solve the inversion problems for velocities only. In our case, when performing the recovery, we simulated the same workflow as that used during the processing of the experimental data; therefore, we faced the same problem in which a trade-off occurs between source and velocity parameters. However, we suggest that our tests more adequately represent the realistic resolution than cases that include fixed sources.

In Supplementary Fig. 7, we present two different checkerboard models with sizes of $50 \times 55$ km and $30 \times 35$ km. The anomalies are defined in the plane of the profile; in the perpendicular direction, the thickness of the anomalies is 50 km to the opposite side of the profile. Based on the results of these tests, we can conclude that the resolution is fair down to a depth of 70 km; however, at greater depths the anomalies are strongly smeared. The capacity for resolving realistic patterns is investigated in another test presented in Supplementary Fig. 8. Here we follow the same strategy as that for the test shown in Supplementary Fig. 6. The anomalies are defined as polygonal prisms in the vertical section, which is the same method used to present the main results in Supplementary Fig. 4. Across the section, the thickness of the prism was 80 km for most patterns and 40 km for anomalies crossing the forearc. When designing the model, we tuned the shapes and amplitudes of anomalies to achieve the maximum resemblance of the recovered anomalies with the S-model presented in Supplementary Fig. 4. This process helped us to estimate realistic amplitudes of anomalies within the structures identified in this vertical section. In particular, for the deeper and shallower reservoirs beneath Toba, this test produced S-wave velocity anomaly values of $-14$ and $-18\%$, respectively. The amplitude of the inclined anomaly connecting the slab at $\sim 80$ km depth with Toba was slightly stronger than the vertical anomaly originating from a depth of $\sim 150$ km. In general, this test indicated that the geodynamic scenario presented in the Discussion section is plausible.

**Data availability.** All of the calculations related to the inversion of the observed data and the performance of the synthetic tests can be reproduced using the full template, including the LOTOS code. All of the data and parameters can be downloaded from www.ivan-art.com/temp/lotos_toba.zip. Following the simple instructions in the user guide, one may obtain all of the results presented in this study.

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

## Acknowledgements

All of the data used in this study were acquired and distributed by the GFZ data centre. I.K. and E.K. are supported by the RNSF grant number 14-17-00430. We appreciate the Deanship of Scientific Research at King Saud University for funding this work through the research group project IRG-14-21. The work of N.S. was supported by the French project 'Labex UnivEarth' and by the Université Sorbonne Paris Cité via project 'VolcanoDynamics'.

## Author contributions

I.K. and E.K. computed the tomography model. I.K. and N.M.S. wrote the majority of the manuscript. E.K., S.E.K. and N.A.A. processed the raw seismic data. S.S., C.J., S.E.K. and N.A.A. provided the major part of the geological interpretation. A.V. was responsible for performing the geoid transformations. All authors contributed to the discussions, interpretation and writing of the paper.

## Additional information

**Competing financial interests:** The authors declare no competing financial interests.

