## [Peer Review File · Nature Communications]

Reviewers' comments:

Reviewer #1 (Remarks to the Author):

A. The manuscript presents a tomographic study of the crust and mantle wedge beneath Toba caldera, Sumatra. The key results presented are new evidence for a dual-level plumbing system with a magma reservoir in the upper crust and a larger reservoir at the base of the crust. I attach two commented files for the main text and additional information. These contain suggestions on how to improve the manuscript. The main comments are listed below as well so there is considerable repetition with the comments written on the manuscript.

B. A previous model had been published in 2009 with data from 1995. Other seismic studies based on different techniques produced images of the crust. The new work presented in the manuscript uses additional data collected in 2008 to produce a better constrained tomographic image of this region and to provide constraints on the deep crust and upper mantle. Given the highly explosive nature of Toba volcanic complex, it is of great importance to understand its magma plumbing system and the model presented could be a valuable contribution. A multi-level system has been postulated for many arc volcanoes based on petrological observations. Current models of magma differentiation predict magma stalling and accumulation at the base of the crust. The fact that the model presented shows evidence for such a multi-level system makes it interesting to a wider audience of volcanologists and petrologists. Unfortunately the importance of this aspect of the model is not pointed out in the paper

C. The paper is generally well written but it is very short. The tomographic inversion seems well carried out, and the data used seems suitable. The density of earthquakes and stations used is better than in previous studies, although still not very high. The deep parts of the model in particular likely suffer from smearing and low resolution that could be more clearly pointed out. This is necessary so that interpretation focuses on the parts of the model that are well constrained. There are some specific issues and omissions that should be addressed:

1. The method is presented in the additional material, which is fine since it is of interest mostly to specialists. However, there should be a more detailed description of the data and tomographic inversion in the main text, so that the reader can understand the novelty, reliability and resolution of the model presented without having to read the highly technical supplement. The main text is very short as it is (under 1400 words), well below the word limit (5000 words).
2. The authors present the model only as plots of velocity anomalies. This is common practice in local earthquake tomography and is appropriate to highlight lateral variations. However, velocity anomalies can be misleading, particularly close to strong geological boundaries (e.g. the Moho). It would help if the authors included at least one plot of absolute velocities, for example for figure S4, where they plot vertical cross-sections.
3. There are no plots of ray coverage or ray density. This is a simple plot that could help better evaluate the model and the potential smearing.
4. I suspect there is an error in the calculation of the variance reduction presented in table 2. A reduction of 43% is not a lot and one would think that the inversion has not been allowed to progress far enough. However the models look detailed enough based on what could be obtained with the given data. I think instead that there may be an error in the variance reduction calculation. The percentages listed are exactly the same as the percentage reduction of RMS residual that can be calculated from the second and third column. This should not be the case since the variance should be of the order of the square of the RMS residual. After this is corrected I suspect that the authors will find that the variance reduction is greater than listed in table 2.

D. Appropriate resolution tests have been carried out and are described in the additional information. As mentioned above the results of these tests should perhaps be briefly discussed in the main text so that the reader can judge which features of the model are robust and can be confidently interpreted. A ray plot coverage or ray density plot could help understand the model robustness.

E. The discussion is extremely synthetic and consists of only one paragraph. Such a short discussion could be appropriate in a technical tomography paper published in a specialized journal, but here all the technical aspects of the paper are hidden away in the additional material. The discussion is the place where the main findings of the tomography model should be discussed in light of previous studies in the regions. Can they help answering any open questions on the functioning of the Toba system and arc volcanoes in general? These are some of the aspects that could be developed:

1. What is the role of the Investigator fracture zone and the corresponding step or tear in the slab?
2. What is the nature of melt transport through the mantle wedge?
3. What are the characteristics of the shallow and deep magma reservoirs that we can deduce from the tomography?
4. Can these observations tell us something new about the deep crustal magma storage zone?
5. There is no discussion of how your model relates to other models tomographic models of volcanoes, apart from a brief mention of the recent paper on Yellowstone.

There is one particular sentence that needs to be discussed or eliminated. At lines 127-128 you say "Certain material from this reservoir may be entrained by the corner flow oriented trenchward (anomaly 2)". This interpretation is based on figure 3D and is also reported in the abstract. The asthenospheric corner flow is unlikely to be able to remove material from the base of the crust unless the lithospheric mantle has been removed by some sort of delamination process or the LAB is particularly shallow because of high temperatures. In general beneath the Moho we expect to find a static lithospheric mantle and the corner flow would not be in contact with the crust. The way you have drawn figure 3D therefore is unrealistic and crucially lacks a lithospheric mantle. In addition anomaly 2 is more likely to be caused by ascending fluids released from the slab between 80 and 100 km depth that would cause serpentinization of the upper corner of the mantle wedge. A plot of absolute velocities (as suggested above) and perhaps of V_p/V_s ratio may help interpretation of these features. In particular the V_p/V_s ratio is expected to be high for serpentinized peridotites.

F. I have listed several other questions and smaller suggestions in the commented files. These are in Microsoft Word docx format.

G. The list of references is quite short. I have suggested a few papers that could provide additional talking points to expand the introduction and discussion. These are listed in the commented files, but I list them here as well for clarity:

1. Costa et al. (2014), *Frontiers of Earth Science*, vol 2, p 16
2. Basset and Watts (2015), *G-cubed*, vol 16(5), p 1508-1540 and 1541-1576
3. Mueller et al. (2008), *G-cubed*, vol 9, doi: 10.1029/2007GC001743
4. Schnabel, et al. (2007), In *AGU Fall Meeting Abstracts*, vol. 1, p. 0560.
5. Larson et al. (1978), *JGR*, vol 83, p 773-782.
6. Annen et al., 2006, *J. Petrology*, v 47, p505

H. The abstract and conclusions don't seem to focus on what this particular work is adding to our view of the Toba caldera. They paint a general picture of magmatic systems at arc volcanoes and why they might produce explosive eruption. This would be fine if it was a review paper, but there is in fact very little discussion of previous work in the text.

In conclusion the manuscript presents a new tomographic study of Toba caldera, which could provide new insight on its magmatic system. Although the tomographic inversion results seem robust (with the caveats listed above) and are interesting, the extremely short discussion fails to describe the results in any detail and to place these results in the context of outstanding open questions on the functioning of Toba and other arc volcanoes.

Reviewer #2 (Remarks to the Author):

This manuscript describes a recent study looking at new seismic data collected near Toba volcano. The key results are the new tomographic model which shows a complete view of the velocity structure beneath the Toba volcano which elucidates the formation of melt from a subducting slab and the storage areas for melt in the crust.

The paper is organized well and is easy to read and is logical and has the appropriate references to supplemental material which helps explain the data and methodology and provides the appropriate resolution tests for a tomographic study.

The conclusions are logical but are a bit lacking and the paper seems to be more of an explanation of observations and is light on the meaning of the observations.

I would like to see a bit more on what these findings mean for the future of the volcano. Things like how this affects the hazard assessment, does this tell us anything new that can help us understand how magma moves through the crust and maybe where we are currently at in the eruption cycle.

A few recommendations on the text are as follows:

line 33: where it states, "it would drastically alter the life of humans for several generations", do you have a reference for this statement?

line 52: I would remove "apparently".

line 73: I would change "supervolcano" to "volcano". The term supervolcano is a reference to the size of an eruption, not the volcano itself. It is true that Toba has produced supervolcanic eruptions, but it has also produced smaller eruptions.

line 105-107: You state, "The strong negative shear velocity anomaly and much weaker P-wave velocity anomaly suggest the presence of significant amounts of melt inside this reservoir". Can you be more quantitative here? Can you estimate the %melt in these reservoirs? You give it a volume of 50,000 cubic km but I'm assuming not all this is melt.

line 110: I would remove the citation 21 here. The actual study is citation 20. Citation 21 was merely an explanation of the Huang et al. study.

Another possibility which may provide more insight is looking at the V_p/V_s ratio to see what that may tell you about the magma reservoirs and the amount of melt involved.

Overall, I think this is a good manuscript but I think it needs just a little bit more to make it more complete.

Reviewer #3 (Remarks to the Author):

This paper presents a seismic tomography model for the Toba magmatic system using natural

seismicity recorded during 1995 and 2008. A recent study by Jaxybulatov, K., et al. (2014) and some of the authors of this paper used the same data to model the shallow (crustal) roots of the Toba system, while this paper focusses on the deep (mantle) source and sub-crustal Toba magma reservoir. Thus, the current paper is a nice follow-up to the Jaxybulatov paper providing a comprehensive model of the Toba magmas from source to sub-volcanic magma chamber for this prolific magma system. It is mostly well-written (see minor comments below) and is well referenced. Major findings in this paper indicate that there is both a seismic and magma generation correlation to the subducted portion of the Investigator Ridge Fracture Zone (IFZ) and that a major sub-crustal magma reservoir exists at the base of the crust directly beneath Toba. These results will be of great interest to those working on subduction zone magma systems and supervolcanoes.

Tomography models, done with both P and S waves and constructed for various depths ranging from 10 to 80 km, were merged with seismic foci into a cross-section that depicts all the major components of a subduction zone magma factory. The algorithm (LOTUS code) and synthetic checkerboard tests seem to be robust to this non-geophysicist. Figures are all necessary and clear while the supplemental text, figures, and table are reasonably appropriate.

While I do believe that the major findings are well-supported by the tomography/seismic models, some of the implications may need to be modified before publication as noted below:

1. In the abstract (line 21) and line 94 the authors state or imply that the melts are being generated in the subducted slab. Conventional thinking of subduction zone magma genesis mostly supports a mantle wedge melt source just above the subducting slab that is triggered by volatiles/fluids from the subducted slab. The authors refer to this process in lines 101-103. Therefore, lines 21 and 94 should be modified to reflect a mantle wedge melt source, which will then be consistent with lines 101-103.

2. In the abstract (line 24), the body (line 111-114), and the conclusions (line 122), the authors indicate that evolved melts (light differentiates) from the mafic magma chamber at the base of the crust then ascend to supply and form the shallow magma reservoir (≤ 10 km) that periodically erupts at Toba. This is a major statement that needs revision based upon geochemical arguments, especially Sr isotope ratios as follows: All of Toba's magmas have $^{87}\text{Sr}/^{86}\text{Sr}$ ratios between about 0.7128 - 0.7152 while typical stratovolcanoes of Sumatra have Sr ratios between about 0.7038-0.7059. This great disparity has been attributed to indicate that the typical Sumatran stratovolcanoes are comprised of mafic melts (and their derivatives) generated from the mantle wedge, while Toba consists almost exclusively of melts derived from continental crust (see Chesner, 2012 for further details). Given the Sr isotopic evidence and implications, the authors should modify these portions of the paper to indicate that the large sub-crustal mafic magma chamber shown by their data (and consisting of mantle-wedge derived melts), provides significant heat (and perhaps volatiles as well) to the overlying continental crust which then melts to produce the intermediate to silicic magmas that rise into the shallow Toba system, evolve further, and eventually erupt.

3. I would temper the statement in line 32 to say that Toba "may have" significantly affected global..... This is because not all researchers believe that Toba had a major influence on the climate (Chesner and Luhr, 2010, among others).

4. The authors refer to Toba as "exceptionally vigorous" (Line 35 and Line 115). This characterization implies to me that it has frequent eruptions; it does not. Instead it would be better to describe it as a system that "periodically has unusually voluminous eruptions."

5. A major premise that the authors use to explain the unusually large volume of mafic magma in the sub-crustal reservoir (50,000 km³) is that the IFZ was especially volatile rich, thus causing anomalous melt production in the overlying mantle wedge. It is a 1500 m high ridge, but why

should it contain more volatiles than adjacent oceanic crust? Perhaps the authors could elaborate in a sentence or two regarding any direct evidence from their cited sources for the water-rich nature of the IFZ (i.e. what is different about the IFZ that causes it to be more volatile rich?).

6. In Lines 104 and 105 the terms "flows" is used to describe the magma "pathways" from their source to the sub-crustal magma chamber. Please use different wording here as "flows" is more suggestive of a surface process whereas this action is one of magma migration.

7. Figure 2 caption should identify by name the other "caldera forms" to the north and south of Toba.

Rebuttal letter with responses to the reviewers on the paper by I. Koulakov et al., "The feeding system of the Toba supervolcano from the slab to the shallow reservoir"
The author's responses are highlighted with violet and start with "REP"

Reviewers' comments:

Reviewer #1 (Remarks to the Author):

A. The manuscript presents a tomographic study of the crust and mantle wedge beneath Toba caldera, Sumatra. The key results presented are new evidence for a dual-level plumbing system with a magma reservoir in the upper crust and a larger reservoir at the base of the crust. I attach two commented files for the main text and additional information. These contain suggestions on how to improve the manuscript. The main comments are listed below as well so there is considerable repetition with the comments written on the manuscript.

B. A previous model had been published in 2009 with data from 1995. Other seismic studies based on different techniques produced images of the crust. The new work presented in the manuscript uses additional data collected in 2008 to produce a better constrained tomographic image of this region and to provide constraints on the deep crust and upper mantle. Given the highly explosive nature of Toba volcanic complex, it is of great importance to understand its magma plumbing system and the model presented could be a valuable contribution. A multi-level system has been postulated for many arc volcanoes based on petrological observations. Current models of magma differentiation predict magma stalling and accumulation at the base of the crust. The fact that the model presented shows evidence for such a multi-level system makes it interesting to a wider audience of volcanologists and petrologists. Unfortunately the importance of this aspect of the model is not pointed out in the paper

REP: The multilayer magma structure beneath Toba and its role in generating large explosive reservoirs in the upper crust is discussed in more detail in the new version of the paper.

C. The paper is generally well written but it is very short. (REP: We have considerably enlarged the paper and included a lot of additional information that appears to be important for the story). The tomographic inversion seems well carried out, and the data used seems suitable. The density of earthquakes and stations used is better than in previous studies, although still not very high. The deep parts of the model in particular likely suffer from smearing and low resolution that could be more clearly pointed out. (REP: We honestly state the problem of limited vertical resolution in L177–179). This is necessary so that interpretation focuses on the parts of the model that are well constrained. There are some specific issues and omissions that should be addressed:
1. The method is presented in the additional material, which is fine since it is of interest mostly to specialists. However, there should be a more detailed description of the data and tomographic inversion in the main text, so that the reader can understand the novelty, reliability and resolution of the model presented without having to read the

highly technical supplement. The main text is very short as it is (under 1400 words), well below the word limit (5000 words).

REP: In the main text, we have included the information about the data (L107–124) and the algorithm (L125–130). More details about performing the tomographic inversion are incorporated into the Method section.

2. The authors present the model only as plots of velocity anomalies. This is common practice in local earthquake tomography and is appropriate to highlight lateral variations. However, velocity anomalies can be misleading, particularly close to strong geological boundaries (e.g. the Moho). It would help if the authors included at least one plot of absolute velocities, for example for figure S4, where they plot vertical cross-sections.

REP: We present the plots of absolute P- and S-velocity in the supplementary material (Figure S4). In our opinion, they do not contain much additional information for our story; therefore, we did not include them in the main paper. Unfortunately, seismic tomography deals only with smooth velocity distributions and cannot reveal any sharp boundaries.

3. There are no plots of ray coverage or ray density. This is a simple plot that could help better evaluate the model and the potential smearing.

REP: We have produced Figure S2 showing the distributions of the ray paths in map view and in vertical section together with the parameterization nodes.

4. I suspect there is an error in the calculation of the variance reduction presented in table 2. A reduction of 43% is not a lot and one would think that the inversion has not been allowed to progress far enough. However the models look detailed enough based on what could be obtained with the given data. I think instead that there may be an error in the variance reduction calculation. The percentages listed are exactly the same as the percentage reduction of RMS residual that can be calculated from the second and third column. This should not be the case since the variance should be of the order of the square of the RMS residual. After this is corrected I suspect that the authors will find that the variance reduction is greater than listed in table 2.

REP: In many tomography papers, authors report reductions of squared norms of the residuals (L2 norm). In our case, if we used this approach, the variance reduction was indeed much stronger: the squared P-residuals reduced from 0.1521 to 0.0655 s^2 (59.91%), and the S-residuals reduced from 0.3745 to 0.1176 s^2 (68.58%). In our study, we prefer leaving residuals in the L1 norm as we did in all our previous papers. However, if the reviewer insists, we will change it in this way as he requests.

D. Appropriate resolution tests have been carried out and are described in the additional information. As mentioned above the results of these tests should perhaps be briefly discussed in the main text so that the reader can judge which features of the model are robust and can be confidently interpreted.

REP: In the main text, we have included more description of the synthetic modeling and pointed out some conclusions (L165–179).

A ray plot coverage or ray density plot could help understand the model robustness.

REP: In the supplementary material, we have included Figure S2 with the distributions of ray paths in horizontal and vertical sections together with the parameterization nodes.

E. The discussion is extremely synthetic and consists of only one paragraph. Such a short discussion could be appropriate in a technical tomography paper published in a specialized journal, but here all the technical aspects of the paper are hidden away in the additional material. (REP: We have considerably expanded the part of the discussion; now it takes three pages). The discussion is the place where the main findings of the tomography model should be discussed in light of previous studies in the regions. Can they help answering any open questions on the functioning of the Toba system and arc volcanoes in general? (REP: In the introduction and discussion sections, we have pointed out three major questions regarding the origin of the subduction-related supervolcanism and discussed them in detail.). These are some of the aspects that could be developed:

1. What is the role of the Investigator fracture zone and the corresponding step or tear in the slab?

REP: Thanks to this small comment, we have considerably changed our view on the role of the IFZ in the origin of the Toba supervolcanism. In particular, we have revisited our regional tomography model beneath the Sunda Arc and found nice evidence for the slab tear beneath Toba. We decided to include this model in our paper as Figure 1C. We have also included Figure 5B showing a schematic representation of the slab tear along the IFZ line and its role in the formation of the multilevel plumbing system beneath Toba. The role of the IFZ is discussed in several places in the paper (L192–202).

2. What is the nature of melt transport through the mantle wedge?

REP: We have added a sentence to specify this issue (L240–241). The nature of the melt transport through the mantle wedge is discussed in L242–245.

3. What are the characteristics of the shallow and deep magma reservoirs that we can deduce from the tomography?

REP: We have added a paragraph discussing the uncertainty in the determination of the amplitudes of anomalies. For the crustal anomalies, we estimate the amplitudes of 16% and 18%. For the deeper anomalies, the uncertainty is much larger and the derived amplitudes should be considered with prudence. The discussion of this issue has been added in L180–189.

4. Can these observations tell us something new about the deep crustal magma storage zone?

REP: In the new version of the paper, we point out that adding the new data allows discrimination of the deeper and shallower reservoirs beneath the Toba Caldera, whereas

in the older model they were identified as a single strongly smeared anomaly. The discussion of this and other issues is presented in L142–160 and L177–179.

5. There is no discussion of how your model relates to other models tomographic models of volcanoes, apart from a brief mention of the recent paper on Yellowstone. REP: We have added a remark that the anomaly at the base of the crust appears very similar to that identified beneath the Klyuchevskoy volcano group in Kamchatka (Koulakov et al., 2011) (see L148–153). We also mention the similarity of seismic structures in the mantle wedge beneath Toba with previously published models in the central and southern Andes (L209–210).

There is one particular sentence that needs to be discussed or eliminated. At lines 127–128 you say "Certain material from this reservoir may be entrained by the corner flow oriented trenchward (anomaly 2)". This interpretation is based on figure 3D and is also reported in the abstract. The asthenospheric corner flow is unlikely to be able to remove material from the base of the crust unless the lithospheric mantle has been removed by some sort of delamination process or the LAB is particularly shallow because of high temperatures. In general beneath the Moho we expect to find a static lithospheric mantle and the corner flow would not be in contact with the crust. The way you have drawn figure 3D therefore is unrealistic and crucially lacks a lithospheric mantle. In addition anomaly 2 is more likely to be caused by ascending fluids released from the slab between 80 and 100 km depth that would cause serpentinization of the upper corner of the mantle wedge. A plot of absolute velocities (as suggested above) and perhaps of Vp/Vs ratio may help interpretation of these features. In particular the Vp/Vs ratio is expected to be high for serpentinized peridotides.

REP: We had a heated discussion among the coauthors about this issue. Finally, we decided to accept the reviewer's point of view, which is also supported by most of us. We have modified the main figure with interpretation (Figure 5A) and changed the text (L203–210). According to this reviewer's comment, we have added the plot of Vp/Vs ratio in the vertical section (Figure 4C).

F. I have listed several other questions and smaller suggestions in the commented files. These are in Microsoft Word docx format.

REP: We have considerably updated the text and included the corrections.

G. The list of references is quite short. I have suggested a few papers that could provide additional talking points to expand the introduction and discussion. These are listed in the commented files, but I list them here as well for clarity:

1. Costa et al. (2014), *Frontiers of Earth Science*, vol 2, p 16
2. Basset and Watts (2015), *G-cubed*, vol 16(5), p 1508-1540 and 1541-1576
3. Mueller et al. (2008), *G-cubed*, vol 9, doi:10.1029/2007GC001743
4. Schnabel, et al. (2007), In *AGU Fall Meeting Abstracts*, vol. 1, p. 0560.

5. Larson et al. (1978), JGR, vol 83, p 773-782.

6. Annen et al., 2006, J. Petrology, v 47, p505

REP: These papers have been carefully considered and most of them are included as references to the paper.

H. The abstract and conclusions don't seem to focus on what this particular work is adding to our view of the Toba caldera. They paint a general picture of magmatic systems at arc volcanoes and why they might produce explosive eruption. This would be fine if it was a review paper, but there is in fact very little discussion of previous work in the text.

REP We have considerably rewritten both the abstract and conclusions.

In conclusion the manuscript presents a new tomographic study of Toba caldera, which could provide new insight on its magmatic system. Although the tomographic inversion results seem robust (with the caveats listed above) and are interesting, the extremely short discussion fails to describe the results in any detail and to place these results in the context of outstanding open questions on the functioning of Toba and other arc volcanoes.

REP: We are grateful to the reviewer for the huge work of reviewing our manuscript and for very friendly and constructive comments. We hope that the reviewer will find our corrected version improved.

Reviewer #2 (Remarks to the Author):

This manuscript describes a recent study looking at new seismic data collected near Toba volcano. The key results are the new tomographic model which shows a complete view of the velocity structure beneath the Toba volcano which elucidates the formation of melt from a subducting slab and the storage areas for melt in the crust.

The paper is organized well and is easy to read and is logical and has the appropriate references to supplemental material which helps explain the data and methodology and provides the appropriate resolution tests for a tomographic study.

The conclusions are logical but are a bit lacking and the paper seems to be more of an explanation of observations and is light on the meaning of the observations.

I would like to see a bit more on what these findings mean for the future of the volcano. Things like how this affects the hazard assessment, does this tell us anything new that can help us understand how magma moves through the crust and maybe where we are currently at in the eruption cycle.

REP: We consider this issue in the final paragraph of the discussion (L275–283).

A few recommendations on the text are as follows:

line 33: where it states, "it would drastically alter the life of humans for several generations", do you have a reference for this statement?

REP: This is an emotional statement rather than a scientific one. It could be removed, but in our opinion, such passages make the paper more attractive for a broad audience. To soften it, we have removed the words "for several generations". In any case, the heavy consequences of a supereruption for humans would be obvious.

line 52: I would remove "apparently".

Done

line 73: I would change "supervolcano" to "volcano". The term supervolcano is a reference to the size of an eruption, not the volcano itself. It is true that Toba has produced supervolcanic eruptions, but it has also produced smaller eruptions.

Replaced to "Caldera".

line 105-107: You state, "The strong negative shear velocity anomaly and much weaker P-wave velocity anomaly suggest the presence of significant amounts of melt inside this reservoir". Can you be more quantitative here? Can you estimate the %melt in these reservoirs? You give it a volume of 50,000 cubic km but I'm assuming not all this is melt.

REP: We have added some discussion on this issue. First, we specify that the reservoir is obviously not liquid. Second, we confess that we cannot exactly retrieve the values of petrophysical parameters due to several types of uncertainties. Third, we provide more explanation about how we estimate the volume of the reservoir (L244–253).

line 110: I would remove the citation 21 here. The actual study is citation 20. Citation 21 was merely an explanation of the Huang et al. study.

REP: This citation has been removed from here and included in another place (L271–272).

Another possibility which may provide more insight is looking at the V_p/V_s ratio to see what that may tell you about the magma reservoirs and the amount of melt involved.

REP: We have included the plot with V_p/V_s ratio (Figure 4C) and added some discussion (L137-138, 206–207 and other places).

Overall, I think this is a good manuscript but I think it needs just a little bit more to make it more complete.

REP: Thank you very much for your friendly comments.

Reviewer #3 (Remarks to the Author):

This paper presents a seismic tomography model for the Toba magmatic system using natural seismicity recorded during 1995 and 2008. A recent study by Jaxybulatov, K., et al. (2014) and some of the authors of this paper used the same data to model the

shallow (crustal) roots of the Toba system, while this paper focusses on the deep (mantle) source and sub-crustal Toba magma reservoir. Thus, the current paper is a nice follow-up to the Jaxybulatov paper providing a comprehensive model of the Toba magmas from source to sub-volcanic magma chamber for this prolific magma system. It is mostly well-written (see minor comments below) and is well referenced. Major findings in this paper indicate that there is both a seismic and magma generation correlation to the subducted portion of the Investigator Ridge Fracture Zone (IFZ) and that a major sub-crustal magma reservoir exists at the base of the crust directly beneath Toba. These results will be of great interest to those working on subduction zone magma systems and supervolcanoes.

Tomography models, done with both P and S waves and constructed for various depths ranging from 10 to 80 km, were merged with seismic foci into a cross-section that depicts all the major components of a subduction zone magma factory. The algorithm (LOTUS code) and synthetic checkerboard tests seem to be robust to this non-geophysicist. Figures are all necessary and clear while the supplemental text, figures, and table are reasonably appropriate.

While I do believe that the major findings are well-supported by the tomography/seismic models, some of the implications may need to be modified before publication as noted below:

1. In the abstract (line 21) and line 94 the authors state or imply that the melts are being generated in the subducted slab. Conventional thinking of subduction zone magma genesis mostly supports a mantle wedge melt source just above the subducting slab that is triggered by volatiles/fluids from the subducted slab. The authors refer to this process in lines 101-103. Therefore, lines 21 and 94 should be modified to reflect a mantle wedge melt source, which will then be consistent with lines 101-103.

REP: In the abstract, we have replaced "A large amount of volatile-enriched melts has been generated in the subducting slab" with "Large amounts of volatiles originate in the subducting slab". In other parts of the paper, the text has been completely rewritten.

2. In the abstract (line 24), the body (line 111-114), and the conclusions (line 122), the authors indicate that evolved melts (light differentiates) from the mafic magma chamber at the base of the crust then ascend to supply and form the shallow magma reservoir (≤ 10 km) that periodically erupts at Toba. This is a major statement that needs revision based upon geochemical arguments, especially Sr isotope ratios as follows: All of Toba's magmas have $^{87}\text{Sr}/^{86}\text{Sr}$ ratios between about 0.7128 - 0.7152 while typical stratovolcanoes of Sumatra have Sr ratios between about 0.7038-0.7059. This great disparity has been attributed to indicate that the typical Sumatran stratovolcanoes are comprised of mafic melts (and their derivatives) generated from the mantle wedge, while Toba consists almost exclusively of melts derived from continental crust (see Chesner, 2012 for further details). Given the Sr isotopic evidence and implications, the authors should modify these portions of the paper to indicate that the large sub-crustal mafic magma chamber shown by their data (and consisting of mantle-wedge derived melts),

provides significant heat (and perhaps volatiles as well) to the overlying continental crust which then melts to produce the intermediate to silicic magmas that rise into the shallow Toba system, evolve further, and eventually erupt.

REP: According to this comment, we have considerably changed our interpretation in the discussion (L256–267), abstract (L23–26), and conclusion (L291–294).

3. I would temper the statement in line 32 to say that Toba "may have" significantly affected global..... This is because not all researchers believe that Toba had a major influence on the climate (Chesner and Luhr, 2010, among others).

REP: We have added a phrase regarding an alternative opinion about the effects of the Toba supereruption (L35–37).

4. The authors refer to Toba as "exceptionally vigorous" (Line 35 and Line 115). This characterization implies to me that it has frequent eruptions; it does not. Instead it would be better to describe it as a system that "periodically has unusually voluminous eruptions."

REP: These phrases have been corrected.

5. A major premise that the authors use to explain the unusually large volume of mafic magma in the sub-crustal reservoir (50,000 km³) is that the IFZ was especially volatile rich, thus causing anomalous melt production in the overlying mantle wedge. It is a 1500 m high ridge, but why should it contain more volatiles than adjacent oceanic crust? Perhaps the authors could elaborate in a sentence or two regarding any direct evidence from their cited sources for the water-rich nature of the IFZ (i.e. what is different about the IFZ that causes it to be more volatile rich?).

REP: We have added more discussion about the effect of the subducting IFZ and emphasize that the excessive volatiles may penetrate to the crust of the IFZ through fractures (L195–202).

6. In Lines 104 and 105 the terms "flows" is used to describe the magma "pathways" from their source to the sub-crustal magma chamber. Please use different wording here as "flows" is more suggestive of a surface process whereas this action is one of magma migration.

REP: In the corrected version, we tried to avoid the word "flow" when talking about magma migration in the mantle.

7. Figure 2 caption should identify by name the other "caldera forms" to the north and south of Toba.

REP: A sentence related to other caldera-related structures has been added to the Figure 2 caption.

REVIEWERS' COMMENTS:

Reviewer #1 (Remarks to the Author):

This is my second review of the manuscript entitled "The feeder system of the Toba supervolcano from the slab to the shallow reservoir".

The authors seem to have taken on board the criticism by myself and the other reviewers and have produced a much improved manuscript. I attach a commented pdf with a few minor comments (use Adobe Reader to see the comments properly).

Best wishes

Reviewer #2 (Remarks to the Author):

This manuscript shows results of a new tomographic study of the Toba volcanic system using data from two recent temporary seismic deployments. Results show a complex magma plumbing system with an upper-crustal magma reservoir, a lower-crustal/upper-mantle magma reservoir, and low-velocity structures associated with subduction.

This is a review of a previously submitted manuscript and the paper has been significantly improved from the previous version. In particular, the discussion and conclusion sections have much more detail and interpretation compared to the initial version which was more of a list of observations.

Although I would prefer the authors to be a little more quantitative in some of their interpretations, I understand that the data may not allow this.

Overall the paper is well written and the analysis is sufficient for the tomographic method including all the necessary resolution tests that must accompany such tomographic inversions.

I only have a few grammatical suggestions:

line 55: Change "geographical" to "geographic" and change "geological" to "geologic".

line 110: change "seismic temporal networks" to "temporary seismic networks".

line 206: I would reference figure 3 after "anomaly "2"".

Like I stated before, this version of the manuscript is much improved and I would recommend it for publication.

Reviewer #3 (Remarks to the Author):

The authors have addressed all of my original comments and significantly modified the petrologic aspects of the mantle/crustal melt sources for Toba. While most of their revisions are more reasonable and in tune with current melt genesis models, there are still some parts of their model that would benefit from further revisions. Please see the suggestions below:

1. In the abstract (line 26-27), discussion (lines 269, 270, and 284), and conclusions (line 297), the authors refer to the crustal rocks that melted to produce the Toba silicic melts as "felsic" and "silicic." These terms are restricted to igneous rocks, and thus imply that the source of the Toba magmas was melting of crustal igneous rocks. The high $^{87}\text{Sr}/^{86}\text{Sr}$ ratios of the Toba rocks require

either a Precambrian igneous source (for which no evidence exists), a metasedimentary source with a significant continental component, or re-melting of younger S-Type granitoid bodies. Gasparon and Varne (1995) characterize Toba and nearby Sumatran granitoids as "S-Type" indicative of a sedimentary source, while Chesner (1998) has also invoked a metasedimentary/metavolcanic source. Thus, the continental crustal source required by the high $87\text{Sr}/86\text{Sr}$ ratios of Toba magmas is likely to have originated either from direct melting of (meta)sediments, or re-melting of S-Type granitoids derived from meta(sedimentary) rocks. Hence, the terms "felsic" and "silicic" should be modified to reflect a primary derivation from metasedimentary rocks.

2. In regards to the Toba silicic magma continental source areas, the authors indicate that only superheated volatiles rise into the continental crust. The basaltic underplating model also allows for intrusion of mafic melts from the sub-crustal chamber into the overlying continental crust, which allows for efficient transfer of heat into the crustal melting zone. To say that there is "no considerable upward migration of melts from the mantle reservoir" (line 263-264), may be an inaccurate oversimplification. Even though the silicic Toba magmas originate from continental crust this doesn't preclude any mafic magmas from entering the source area, which may be far more important and plausible than simply transferring heat via rising volatiles.

3. The authors set out to answer 3 main questions about the Toba magmatic system. Their second question "Why were the super-eruptions followed by long periods of quiescence?" The periodicity in the proposed model isn't apparent, and to this reader, the question remained unanswered.

4. The "avalanche-type" process that leads to eruption is confusing. Would the term "chain-reaction" be better?

5. Minor suggestions include:

a) Line 38 - add "in Southeast Asia" after "human life"

b) Line 286 - unclear...do you mean 10's of thousands to 100's of thousands or 10's of years to 100's of thousands of years? Please clarify.

c) Line 302 - replace "volcano" with "system" or "caldera"

d) Figure 2B - add a "+" sign for Pardepur Island in southern Lake Toba

e) Figure 3 - the "violet dotted line" referred to in the text (Line 255) doesn't show up as "dotted" on my image, nor does it appear as "violet".

Rebuttal letter on the revision of the paper by Koulikov et al. "The feeder system of the Toba supervolcano from the slab to the shallow reservoir"

The author's responses are highlighted with violet color and indicated with REP

Reviewer #1 (Remarks to the Author):

This is my second review of the manuscript entitled "The feeder system of the Toba supervolcano from the slab to the shallow reservoir".

The authors seem to have taken on board the criticism by myself and the other reviewers and have produced a much improved manuscript. I attach a commented pdf with a few minor comments (use Adobe Reader to see the comments properly).

Best wishes

REP: Thank you very much for careful revision. We have made a few minor corrections according to the reviewer's suggestions, such as:

L35: We have replaced "huge" with "enormous". We think that it is important to emphasize here and exceptionally large amount of ejected material.

L199-203: To ground the statements, we have added the reference to Maruyama and Okamoto (2007) who presented nice illustrations how water may penetrate to the fracture zone and be transported to subduction zones.

L297-298. We have added "is spasmodically emptied during recurrent but sporadic catastrophic eruptions".

Figure 2: "blue" is replaced with "red".

Reviewer #2 (Remarks to the Author):

This manuscript shows results of a new tomographic study of the Toba volcanic system using data from two recent temporary seismic deployments. Results show a complex magma plumbing system with an upper-crustal magma reservoir, a lower-crustal/upper-mantle magma reservoir, and low-velocity structures associated with subduction.

This is a review of a previously submitted manuscript and the paper has been significantly improved from the previous version. In particular, the discussion and conclusion sections have much more detail and interpretation compared to the initial version which was more of a list of observations.

Although I would prefer the authors to be a little more quantitative in some of their interpretations, I understand that the data may not allow this.

REP: We honestly state that seismic tomography does not provide exact numerical values that could be used for quantitative interpretation. Unfortunately, this is a common problem of such studies.

Overall the paper is well written and the analysis is sufficient for the tomographic method including all the necessary resolution tests that must accompany such tomographic inversions.

I only have a few grammatical suggestions:

line 55: Change "geographical" to "geographic" and change "geological" to "geologic". REP: Corrected

line 110: change "seismic temporal networks" to "temporary seismic networks". REP: Corrected

line 206: I would reference figure 3 after "anomaly 2".

REP: We have added the reference to the two left panels in Figure 4.

Like I stated before, this version of the manuscript is much improved and I would recommend it for publication.

Reviewer #3 (Remarks to the Author):

The authors have addressed all of my original comments and significantly modified the petrologic aspects of the mantle/crustal melt sources for Toba. While most of their revisions are more reasonable and in tune with current melt genesis models, there are still some parts of their model that would benefit from further revisions. Please see the suggestions below:

1. In the abstract (line 26-27), discussion (lines 269, 270, and 284), and conclusions (line 297), the authors refer to the crustal rocks that melted to produce the Toba silicic melts as "felsic" and "silicic." These terms are restricted to igneous rocks, and thus imply that the source of the Toba magmas was melting of crustal igneous rocks. The high $^{87}\text{Sr}/^{86}\text{Sr}$ ratios of the Toba rocks require either a Precambrian igneous source (for which no evidence exists), a metasedimentary source with a significant continental component, or re-melting of younger S-Type granitoid bodies. Gasparon and Varne (1995) characterize Toba and nearby Sumatran granitoids as "S-Type" indicative of a sedimentary source, while Chesner (1998) has also invoked a metasedimentary/metavolcanic source. Thus, the continental crustal source required by the high $^{87}\text{Sr}/^{86}\text{Sr}$ ratios of Toba magmas is likely to have originated either from direct melting of (meta)sediments, or re-melting of S-Type granitoids derived from meta(sedimentary) rocks. Hence, the terms "felsic" and "silicic" should be modified to reflect a primary derivation from metasedimentary rocks.

REP: In this paper, we cannot analyze the details of geochemical and petrophysical processes because we do not have enough data for this. Therefore, throughout the text, we have replaced the words "felsic" and "silicic" with more general "upper crust rocks".

2. In regards to the Toba silicic magma continental source areas, the authors indicate that only superheated volatiles rise into the continental crust. The basaltic underplating model also allows for intrusion of mafic melts from the sub-crustal chamber into the overlying continental crust, which allows for efficient transfer of heat into the crustal melting zone. To say that there is "no considerable upward migration of melts from the mantle reservoir" (line 263-264), may be an inaccurate oversimplification. Even though the silicic Toba magmas originate from continental crust this doesn't preclude any mafic magmas from entering the source area, which may be far more important and plausible than simply transferring heat via rising volatiles.

REP: We have slightly softened our statement. In L258-260, we state that in other volcanoes of Sumatra, there is an important mafic component, which means that melts from the mantle sources are indeed capable reach the surface. At the same time, for the Toba Caldera, "there is no considerable contribution of melts from the mantle in forming the upper crust

reservoir". "No considerable contribution" means that we do not completely exclude the ascent of the mafic magma, but its amount is relatively small. This seems to be absolutely consistent with the reviewer's remark.

3. The authors set out to answer 3 main questions about the Toba magmatic system. Their second question "Why were the super-eruptions followed by long periods of quiescence?" The periodicity in the proposed model isn't apparent, and to this reader, the question remained unanswered.

REP: In L272-274, we have added a sentence specifying this issue: "The accumulation of a sufficient amount of "explosives" in the upper crust takes hundreds of thousands years, which may give the answer to the second question related to long periods of quiescence between supereruptions."

4. The "avalanche-type" process that leads to eruption is confusing. Would the term "chain-reaction" be better?

REP: We would prefer using the "avalanche-type process" because at each next step, the processes become more and more intensive. A chain-reaction presumes similar scale of processes and does not necessarily brings to a catastrophe.

5. Minor suggestions include:

a) Line 38 - add "in Southeast Asia" after "human life"

REP: In our opinion, if a supereruption occurs, it would alter the human life globally, not only in Southeastern Asia.

b) Line 286 - unclear...do you mean 10's of thousands to 100's of thousands or 10's of years to 100's of thousands of years? Please clarify.

REP: It is corrected as: "dozens of thousand or hundreds of thousand".

c) Line 302 - replace "volcano" with "system" or "caldera"

REP: Replaced with "volcano system".

d) Figure 2B - add a "+" sign for Pardepur Island in southern Lake Toba

REP: We have added this volcano to this and other figures of the main paper and supplementary.

e) Figure 3 - the "violet dotted line" referred to in the text (Line 255) doesn't show up as "dotted" on my image, nor does it appear as "violet".

REP: We have increased the thickness of the violet line so that it should be better visible.